# Position: Generative Models Erode Human Temporal Learning Through Market Selection

**Wenjun Cao** [1]

## Abstract

We argue that modern generative models create structural risks for knowledge and cultural production at current, sub-AGI capability levels. We define *Human Temporal Learning* (HTL) as path-dependent knowledge accumulation through sustained engagement with problems over time. Generative outputs increasingly resemble HTL-intensive work in surface features, so verifying whether a given output reflects genuine human learning grows costly relative to its expected benefit. Once verification loses economic justification, evaluators reward outputs regardless of production mode, and producers who invested years of learning compete on price against outputs that cost almost nothing to generate. We call this pathway *value collapse* and formalize it through a costly-inspection framework. Cross-domain evidence from academic publishing, legal practice, content platforms, and software security maps onto four stages of verification erosion. Alignment success is orthogonal. Better-aligned models narrow observable gaps between human and AI outputs, making source verification harder and intensifying competitive pressure against HTL-intensive work even when individual AI outputs improve.

## 1. Introduction

Much AI risk literature focuses on AGI loss of control (Bostrom, 2014; Russell, 2019; Hendrycks et al., 2025). We ask a nearer question: before those thresholds, does machine learning already create structural risk for knowledge and cultural production?

We use the term *Human Temporal Learning* (HTL) for path-dependent knowledge accumulation through sustained engagement with problems over time (Polanyi, 1966). Extended engagement produces judgment and skill that resist codification. Historically, outputs served as signals of quality because producing them required sustained learning, and institutions rewarded this embedded time investment (Spence, 1973).

**Generative models create structural risks to knowledge and cultural production by lowering observable distinctions between deep human work and AI-generated outputs, making verification costly relative to its expected benefit, pushing institutions toward undifferentiated evaluation, and creating competitive pressure against producers who invest in sustained human learning.** We call the resulting pathway *value collapse*. It operates at current sub-AGI capability levels through ordinary market dynamics and is orthogonal to alignment success. As AI outputs improve, the observable gap between human and AI work narrows, making verification harder and intensifying displacement even when individual AI outputs are locally beneficial. Evidence from academic publishing, legal and clinical practice, content platforms, and software security suggests erosion is already underway at varying rates across domains (Liang et al., 2025; Suchak et al., 2025; Esau et al., 2025; Brynjolfsson et al., 2025a).

**Contributions.** (i) A framework showing how the growing difficulty of distinguishing AI-generated from human-produced outputs leads to competitive displacement of deep human work through ordinary market dynamics. (ii) A four-stage classification that organizes cross-domain evidence by how far verification has eroded in each domain. (iii) Governance recommendations targeting the conditions that drive verification erosion to preserve the economic viability of sustained human learning.

## 2. Human Temporal Learning

*Human Temporal Learning* (HTL) is how understanding develops through repeated engagement with problems over time.[1] Human understanding is inherently temporal, trans-

---

[1]Independent Researcher. Correspondence to: Wenjun Cao <wenjun.cao.research@gmail.com>.

*Proceedings of the 43rd International Conference on Machine Learning*, Seoul, South Korea. PMLR 306, 2026. Copyright 2026 by the author(s).

---

[1]Unrelated to temporal sequence modeling in machine learning.

forming as one revisits earlier experience and encounters new aspects of familiar problems (Husserl, 1954; Heidegger, 1927). Extended engagement builds judgment and skill that resist codification (Polanyi, 1966).

HTL once carried direct economic weight. Outputs such as research papers and creative works condensed long learning trajectories, and producing them required sustained engagement. Institutions used this time investment as a proxy for quality (Spence, 1973). Funding agencies favored researchers with extended publication records, journals weighted demonstrated expertise, and hiring committees relied on years of training as evidence of deep competence. Generative models disrupt this arrangement by producing outputs that resemble HTL-intensive work in surface features without the underlying learning process.

**Generative Models Erase Learning Traces.** Machine learning extracts patterns from human artifacts such as papers, code, and creative works. Training optimizes for matching observed outputs, and the learning trajectory behind those outputs drops out of the process entirely. After expensive pretraining, generative models (Vaswani et al., 2017; Ho et al., 2020; Lipman et al., 2023) produce plausible outputs cheaply and at scale (Brown et al., 2020; OpenAI et al., 2024a; DeepSeek-AI et al., 2025).

Formatting, rhetorical structure, citation patterns, and stylistic coherence become increasingly easy to replicate. Distinguishing whether a given output reflects genuine human learning requires going beyond surface assessment to check citations against real sources, audit methodological choices, or test whether reasoning holds under scrutiny. Individual-level detection remains difficult even after extensive research (Liang et al., 2025). As generative outputs grow more polished, surface-level checks lose their ability to distinguish production modes. Evaluators who want the same level of assurance must invest in deeper inspection, raising per-item costs.

## 3. Market Selection

Can evaluators economically justify distinguishing HTL-intensive work from low-HTL output? When checking costs more than the expected benefit, evaluators stop trying. Once they stop, rewards become blind to whether the work involved sustained human learning. Producers who invested years of learning compete on price against outputs that cost almost nothing to generate. High-cost producers exit, the pool shifts toward low-HTL output, and the cycle reinforces itself. We call this pathway *value collapse*.

### 3.1. Setup

For tractability, we compress the spectrum of production modes into two types. Both types may use AI tools, differ-

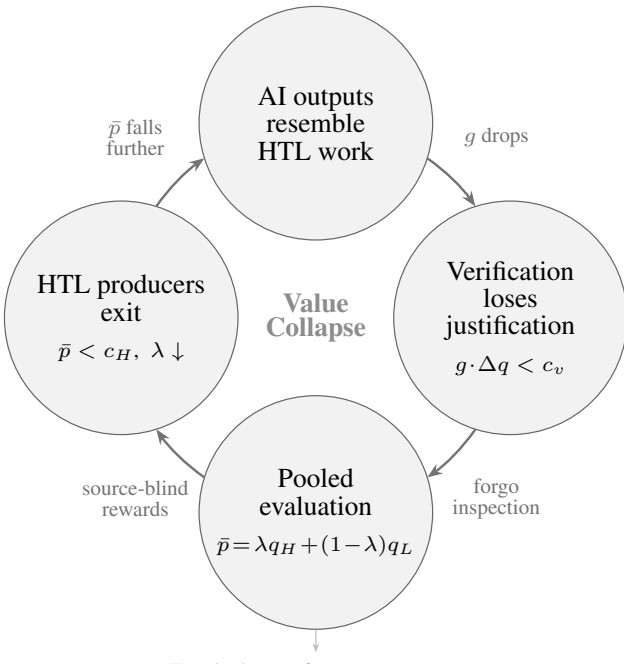

*Figure 1.* Value collapse feedback loop. $g$ = verification ability, $\Delta q$ = quality gap between HTL-intensive and low-HTL output, $c_v$ = per-item verification cost, $\lambda$ = HTL share of the output pool, $\bar{p}$ = pooled reward, $c_H$ = HTL production cost. Generative outputs lower $g$. Once $g \cdot \Delta q < c_v$, evaluators forgo inspection and rewards pool across production types. HTL-intensive producers exit, $\lambda$ falls, and pooled reward declines further.

ing in irreducible human temporal investment. In practice, production lies on a continuum, but the two-type formalization captures the economic logic while keeping the mechanism transparent. We build on models of quality uncertainty and signaling (Akerlof, 1970; Rothschild & Stiglitz, 1976; Spence, 1973). A formal treatment appears in Appendix A.

Four quantities govern whether evaluators will invest in distinguishing the two types.

**Definition 3.1** (Verification ability). How reliably available inspection can distinguish HTL-intensive output from low-HTL output. We denote it $g$. A value of zero means no feasible method separates the two, and a value of one means inspection identifies the source exactly. Detection failures and audit outcomes can serve as empirical proxies for $g$. Institutional responses reflect the broader verification condition, including cost structures and quality stakes.

**Definition 3.2** (Verification cost). The per-item cost of deep inspection, denoted $c_v$. Includes expert time to check citations, audit methods, trace reasoning, or test whether outputs reflect genuine understanding.

**Definition 3.3** (Quality gap). Let $q_H$ and $q_L$ denote the expected payoff-equivalent values of HTL-intensive and low-HTL outputs, measured in the same units as verification

cost. The quality gap $\Delta q = q_H - q_L > 0$ captures the payoff stakes of correct classification for a single output.

**Definition 3.4** (HTL share). The fraction of outputs in the pool that involve genuine human temporal learning, denoted $\lambda$.

### 3.2. Verification Threshold

Verification is worthwhile when the expected gain from distinguishing production types exceeds the cost of trying. Verification ability and quality gap jointly determine whether inspection pays off. Verification is justified when

$$g \cdot \Delta q \gtrsim c_v. \tag{1}$$

When the product of verification ability and quality gap falls below verification cost, evaluators cannot justify deep inspection. Rearranging gives a threshold

$$g^* \approx \frac{c_v}{\Delta q}, \tag{2}$$

where pool composition is held fixed. Below this threshold, rational evaluators forgo verification.

### 3.3. Value Collapse

Once evaluators stop distinguishing, rewards track the average quality of the pool. If deep human work makes up a fraction $\lambda$ of the pool, the pooled reward is

$$\bar{p} = \lambda q_H + (1 - \lambda) q_L. \tag{3}$$

In competitive-price settings $\bar{p}$ is the average price across the pool. In non-price institutions such as journals or platforms it represents the expected reward assigned under source-blind evaluation. More HTL-intensive work raises the average, more low-HTL work lowers it.

When pooled reward covers AI generation costs but falls short of deep human work costs, producers who invest in sustained learning can no longer break even. Deep human work exits. As it exits, the HTL share drops and pooled reward falls further. With heterogeneous costs among HTL-intensive producers, marginal producers exit first, and the cycle continues (Akerlof, 1970). Markets continue producing, but the share reflecting genuine human temporal learning declines.

Once HTL share has fallen far enough, the expected gain from inspection itself declines, reinforcing the shift toward undifferentiated evaluation. This dynamic, in which inability to distinguish quality causes high-cost producers to exit and pool quality to decline, is known in economics as adverse selection.

## 4. Four Stages of Verification Erosion

We organize cross-domain evidence into four stages, ordered by how far verification has eroded in each domain.

### 4.1. Stage 1: Intact Verification

In clinical medicine, patient safety creates quality stakes high enough that physician review of AI-generated output persists. Recent systems assist with patient-friendly discharge summaries, hospital-course summaries, and emergency department documentation (Zaretsky et al., 2024; Small et al., 2025; Song et al., 2025). In each case, clinician oversight remains part of the workflow. AI enters clinical documentation, yet the evaluation process has not collapsed into source-blind acceptance.

### 4.2. Stage 2: Sanctions Maintain Verification

In legal practice, erroneous filings trigger sanctions, malpractice exposure, and reputational losses that together create quality stakes large enough for verification to persist even as its costs rise. Courts have sanctioned attorneys for submitting AI-fabricated citations (United States District Court for the Southern District of New York, 2023) and imposed standing orders requiring attorneys to certify AI usage (United States District Court for the Eastern District of Pennsylvania, 2023). Sanctions, public reprimand, and mandatory training carry real consequences and change behavior. Attorneys now check AI-generated briefs before filing because the cost of failing to check exceeds the cost of checking. Verification holds in this domain because institutions have made the penalty for skipping it severe enough to justify the expense, even though verification itself is costly.

### 4.3. Stage 3: Volume Overwhelms Verification

More broadly, LLM-modified content in computer science abstracts has reached an estimated 22.5%, up from 2.4% before ChatGPT (Liang et al., 2025). For nearly a decade, papers analyzing single-factor statistical associations using the NHANES health database appeared at a steady rate. Around the period of widespread LLM availability, publication rates surged approximately 47-fold (Suchak et al., 2025). A follow-up study found hundreds of papers analyzing identical exposure-outcome pairs, with redundancy increasing seventeen-fold between 2022 and 2024 (Maupin et al., 2025). Each additional redundant paper is likely to contribute little new knowledge, yet still consumes scarce peer-review resources. When marginal contribution is low and per-paper review costs remain substantial, the bar for justified verification rises beyond what the system can sustain.

At top venues, quality stakes are higher, yet verification is still failing under volume pressure. A field report on

| **Stage 1** Intact | **Stage 2** Sanctions | **Stage 3** Overwhelmed | **Stage 4** Source-blind |
|---|---|---|---|
| $g \cdot \Delta q \gg c_v$ | $g \cdot \Delta q \geq c_v$ | $g \cdot \Delta q < c_v$ | $\Delta q \to 0$ |
| High $\Delta q$ sustains verification | Penalties keep $\Delta q$ large | Volume raises $c_v$, lowers $g$ | Evaluation ignores source |
| **Clinical medicine** | **Legal practice** | **Academic publishing** | **Content platforms** |

Verification erosion

*Figure 2.* Four stages of verification erosion and corresponding parameters. $g$ = verification ability, $\Delta q$ = quality gap, $c_v$ = verification cost. Domains are ordered by how far verification has eroded, from intact clinical oversight (Stage 1) to source-blind platform rewards (Stage 4).

ICLR 2026 submissions found at least 50 out of 300 examined papers containing fabricated citations that had received 3–5 expert peer reviews without detection (Esau et al., 2025). Reviewers could in principle verify every reference, but doing so at production volume is infeasible given that global peer-review labor was estimated to exceed 100 million hours in 2020 (Aczel et al., 2021).

A second channel operates through review capacity directly. In the cURL open-source security project, AI-generated vulnerability reports accounted for about 20% of 2025 submissions while the overall valid-report rate fell to roughly 5%, leading the maintainer to consider dropping monetary rewards or restructuring the program (Stenberg, 2025).

### 4.4. Stage 4: Source-Blind Rewards

Digital content platforms allocate rewards through engagement metrics such as watch time, clicks, and shares. When two outputs attract the same attention, the reward mechanism does not ask how they were produced.

Several platforms have introduced AI content labeling. YouTube requires disclosure of realistic AI-generated or meaningfully AI-altered content (YouTube, 2024), Meta applies labels to AI-generated material (Meta, 2025), TikTok has adopted automated labeling and additional transparency tools (TikTok, 2024; 2025), and Spotify has established protections for artists (Spotify, 2025). The EU is developing a code of practice for marking AI-generated content (European Commission, AI Office, 2025). Labels inform viewers. Available evidence suggests that disclosure alone does not systematically alter how platforms distribute rewards. YouTube states that disclosure does not limit audience reach or monetization eligibility, and Spotify states that responsible AI disclosure is not a basis for downranking. Absent broader provenance-sensitive ranking or monetization rules, disclosed AI content faces no automatic distribution or monetization penalty relative to comparable human-created content, leaving the pooled reward structure largely intact.

Generative AI content markets have experienced rapid growth (Grand View Research, 2025; Research and Markets, 2025). Scale economies in foundation models and concentrated platform power create competitive pressure against human-intensive alternatives (Vipra & Korinek, 2023).

## 5. Discussion

### 5.1. Pipeline Compression

Even where verification holds, a subtler threat operates through the training pipeline. If AI automates entry-level tasks, the experiential path that builds senior judgment may narrow while final outputs remain screened. Early-career workers in AI-exposed occupations show a 16% relative employment decline, while employment for experienced workers remained stable (Brynjolfsson et al., 2025a). Big Tech new-graduate hires declined 25% from 2023 to 2024 (SignalFire, 2025). Entry-level task automation connects to broader accounts of displacement and task reallocation (Brynjolfsson et al., 2025a; Hazra et al., 2025; Acemoglu & Restrepo, 2019).

Current stability in high-stakes domains may mask long-run erosion of the HTL pipeline that sustains verification capacity itself. If fewer juniors gain deep experience, the evaluator pool contracts, eventually raising per-item verification costs and pushing the verification threshold upward even in domains where verification currently holds.

### 5.2. From Value Collapse to Model Collapse

As described in Section 3.3, value collapse is the progressive exit of HTL-intensive producers when undifferentiated evaluation fails to reward their higher costs. A related failure mode is *model collapse*, where repeated training on model-generated data degrades distributional coverage (Shumailov

et al., 2024; Alemohammad et al., 2024; Schaeffer et al., 2025). Value collapse increases exposure to model collapse by shifting pool composition. As economic dynamics push HTL-intensive producers out of the pool, training datasets for next-generation models increasingly contain outputs from current-generation models, eroding the distributional diversity that sustained earlier generations.

### 5.3. Pre-AGI Risk

Much AI risk literature emphasizes capability thresholds, containment failures, and governance challenges for highly advanced systems (Bostrom, 2014; Tegmark, 2017; Russell, 2019; Hendrycks et al., 2025; Slattery et al., 2026; Bengio et al., 2024). Value collapse operates on a different axis, concerning present-day deployments at sub-AGI capability levels, and activates when verification can no longer reliably distinguish human from AI-generated outputs at justifiable cost, determined by statistical similarity and cost structures independently of absolute capability levels. Empirical evidence suggests this threshold has already been crossed in important domains.

Value collapse complements takeover-focused perspectives. If societies lose capacity for sustained HTL, future generations have a thinner reservoir of human competence for addressing subsequent risks. Erosion of HTL capacity today constrains society's ability to govern more advanced systems tomorrow and to maintain robust alternatives when AI-dependent infrastructure fails (Kulveit et al., 2025). Value collapse operates without requiring control failures, intent misalignment, or deceptive behavior. Markets selecting for low-cost production, platforms optimizing engagement, and institutions making rational decisions under resource constraints prove sufficient.

### 5.4. Alignment Is Orthogonal

Alignment techniques aim to make AI systems more helpful, harmless, and honest (Ziegler et al., 2020; Ouyang et al., 2022; Bai et al., 2022a;b; OpenAI et al., 2024a), achieving substantial and well-documented success (OpenAI et al., 2024b;c; Perez et al., 2023; Sharma et al., 2024). Recent work examines structural limits on this process (Cao, 2025a). Relative to value collapse, alignment is largely orthogonal. Greater alignment increases model usability, raising adoption rates and expanding tasks users delegate to AI assistance. In output-based verification settings, alignment reduces visible separability, because models that avoid obvious errors, follow formatting conventions, and use citations more reliably become harder to distinguish from careful human work through surface checks alone. Where verification relies on process records, output alignment has no necessary downward effect on verification ability, and provenance-oriented alignment tools could even strengthen verification

by making production histories easier to audit.

Because many high-volume evaluation institutions primarily assess outputs rather than production processes, output similarity currently affects more settings than provenance-based verification can reach. When AI outputs become harder to distinguish from human work, source verification becomes even less economically justifiable, shifting evaluation toward source-blind reward allocation. Selection pressure against HTL-intensive work can therefore strengthen even when individual AI outputs improve, in a dynamic resembling reward-model overoptimization (Gao et al., 2023). Value collapse remains compatible with models behaving in locally aligned ways (Kulveit et al., 2025; Bengio et al., 2024). Alignment work remains essential, yet market-level adverse selection requires complementary institutional design. Alignment efforts incorporating process transparency, reasoning traces, or provenance disclosure could partially counteract the output-similarity channel by making production histories auditable.

### 5.5. Limitations

Domains differ in how they value long training, control provenance, and balance openness with protecting HTL-embedded work. Open-science institutions and reputational systems can support long-term investment (Dasgupta & David, 1994), yet exposing deeply embedded social capacities to unbounded market logics can erode protective structures (Polanyi, 1944). Productivity gains from AI assistance (Noy & Zhang, 2023; Dell'Acqua et al., 2023; Chen et al., 2025; Eloundou et al., 2024; Chatterji et al., 2025) represent a potentially mitigating factor, though aggregate quality indicators in academic publishing show concerning trends even as individual researchers become more productive (Suchak et al., 2025; Maupin et al., 2025). Selection pressure operates continuously through normal market mechanisms while institutional change proceeds through slow deliberation.

## 6. Alternative Views

**Productivity gains will offset displacement.** Substantial productivity evidence exists (Noy & Zhang, 2023; Brynjolfsson et al., 2025b; Gilardi et al., 2023). Individual-level gains do not prevent systemic value collapse when verification erodes. NHANES publication volume surged even as individual researchers became more productive, yet review systems could not maintain quality at the resulting scale. Early-career workers in AI-exposed occupations face a 16% relative employment decline while employment for experienced workers in the same occupations remains stable (Brynjolfsson et al., 2025a), and displacement concentrates precisely where HTL investment is still accumulating. Automation theory shows that productivity gains can coexist with displacement when new tasks do not fully offset auto-

mated ones (Acemoglu & Restrepo, 2019). Value collapse is the analogous risk when institutions fail to preserve source-sensitive rewards for HTL.

**Quality differences remain detectable by experts.** Deep verification may be feasible for any single output. Feasibility for individual items does not make verification economically worthwhile at production volume. Expert evaluation requires significant time from highly qualified specialists, and global peer-review labor is already strained at scale (Aczel et al., 2021). Fabricated citations in ICLR 2026 submissions passed multiple expert reviewers who could in principle have caught them through careful reference checking. Our framework requires only that verification loses economic justification at the relevant volume. Evidence showing experts routinely and economically distinguishing sources at production scale would challenge the four-stage ordering.

**Institutions will adapt to maintain quality.** By late 2025, most major publishers had established ethical frameworks requiring AI disclosure (Gewaltig, 2025; Zhu, 2025), courts had imposed sanctions and standing orders, and platforms had mandated AI content labels. Fabricated citations continue to pass expert review, and formulaic papers enter the literature at unprecedented volume alongside disclosure policies (Suchak et al., 2025; Esau et al., 2025). Enforcement proves difficult when compliance is voluntary and competitive incentives favor non-disclosure. Market dynamics do not pause while governance catches up.

**New provenance technologies may restore separation.** Watermarking and detection tools (Kirchenbauer et al., 2023; Zhao et al., 2024; Wen et al., 2023) can raise verification ability and reduce per-item inspection costs, and should be pursued. Adversarial dynamics of detection remain challenging (Sadasivan et al., 2025), adoption requires coordination across fragmented markets, and competitive pressures create incentives for circumvention. Whether deployment and enforcement can sustain improved verification against continuous downward pressure remains an open empirical question.

**Quality convergence justifies displacement.** If AI outputs truly match HTL quality in a domain, displacement of high-cost production may represent efficient reallocation rather than market failure. We acknowledge this possibility for terminal-consumption goods where quality convergence is genuine. Our analysis concerns domains where quality differences remain socially meaningful because deep human learning matters for downstream research, training data quality, evaluator capacity, and novel problem-solving. In these domains, market-level adverse selection operates even when the quality gap is real and consequential.

**HTL-intensive work may survive as a premium niche.** Industrialization displaced artisan weaving. Handwoven textiles survived as a high-end niche. HTL-intensive work may follow a similar path, retreating to a premium segment rather than disappearing. Textiles, however, are terminal consumption goods, and machine-produced cloth does not re-enter the loom's design process. Knowledge serves as a production factor for subsequent knowledge. Research produced today shapes future research and supplies training data for next-generation models. If HTL retreats to a niche, training data diversity declines, the evaluator pool contracts, and AI-generated content re-enters training corpora in a recursive feedback loop with no analogue in physical manufacturing. Producers who choose low-HTL production do not bear these costs. Degraded training data, a shrinking evaluator pool, and reduced capacity for novel problem-solving fall on future researchers, evaluators, and downstream users. Because the costs are diffuse and deferred, source-blind markets lack strong mechanisms to make producers bear them.

## 7. Related Work

**Philosophical Foundations of Temporal Learning.** Phenomenology provides foundational analyses of lived time and world-embedded experience (Husserl, 1954; Heidegger, 1927), while work on technology examines how technical systems reorganize human temporality (Heidegger, 1954; Wiener, 1950). Tacit knowledge theory (Polanyi, 1966) establishes that extended engagement builds judgment resisting codification. Economic anthropology (Polanyi, 1944) examines how market expansion can erode social capacities that markets themselves depend on. Our operational notion of HTL draws on these analyses and gives the connection between temporal engagement and tacit judgment a specific economic function, connecting the irreducibility of temporal learning to the adverse-selection mechanism through which generative models displace HTL-intensive production.

**Economics of Quality Uncertainty and Scientific Institutions.** Our costly-inspection framework builds directly on Akerlof's quality-uncertainty model (Akerlof, 1970) and related work on adverse selection (Rothschild & Stiglitz, 1976) and signaling, where costly actions such as extended training reveal private information about quality (Spence, 1973). Gresham's Law (Mundell, 1998; Rolnick & Weber, 1986), the principle that debased currency displaces sound currency when both circulate at equal face value, provides a historical analogy. Our framework operates through information asymmetry alone, where evaluators cannot reliably distinguish production types at justifiable cost. Research on economics of science (Dasgupta & David, 1994) studies how priority rules and institutional designs support long-term investment, while work on platform economics

(Vipra & Korinek, 2023) examines how concentrated platform power reshapes competitive dynamics. AI systems can also be understood as markets in which producers lose value when platforms become endpoints (Jordan, 2025), an analysis complementary to our focus on verification erosion. Prior work (Cao, 2025b) formalizes how opaque AI architectures internalize user contributions, identifying a structural risk distinct from the adverse-selection mechanism analyzed here.

**AI Productivity and Labor Market Effects.** Experiments demonstrate substantial individual-level productivity gains from LLM assistance (Noy & Zhang, 2023; Brynjolfsson et al., 2025b), with evidence of task reallocation across skill boundaries (Dell'Acqua et al., 2023) and documented adoption patterns across occupations and countries (Eloundou et al., 2024; Chatterji et al., 2025). Employment analyses reveal heterogeneous impacts, with early-career workers facing relative decline while employment for experienced workers remained stable (Brynjolfsson et al., 2025a; SignalFire, 2025). Broader analyses of automation and task displacement (Acemoglu & Restrepo, 2019; Hazra et al., 2025) examine how differential displacement across experience levels can narrow the pipeline through which expertise develops, a dynamic central to pipeline compression.

**Verification Erosion across Domains.** Large-scale studies quantify rising LLM usage in scientific papers (Liang et al., 2025) and document concerning publication trends including formulaic surges (Suchak et al., 2025) and rapidly growing redundancy (Maupin et al., 2025). Field reports reveal fabricated citations passing expert peer review (Esau et al., 2025), AI-generated submissions overwhelming open-source security review (Stenberg, 2025), and AI-assisted contributions receiving lighter review scrutiny in open-source projects (Gao et al., 2026). Automating peer review without adequate safeguards introduces further risks (Baumann et al., 2026), and automated agents that inflate submission volumes present a systemic challenge to conference evaluation capacity (Shan et al., 2026). In legal and medical domains, judicial sanctions for AI fabrications (United States District Court for the Southern District of New York, 2023; United States District Court for the Eastern District of Pennsylvania, 2023) and clinical AI documentation studies (Zaretsky et al., 2024; Small et al., 2025; Song et al., 2025) illustrate verification dynamics at different stages. Our four-stage framework organizes this cross-domain evidence within a unified economic logic, explaining why domains differ in vulnerability to verification erosion.

**Advanced AI Risks and Alignment.** Work on highly advanced systems examines capability thresholds, loss of control, and governance challenges for AGI (Hendrycks et al., 2025; Bostrom, 2014; Russell, 2019). Particularly relevant

is work on gradual disempowerment driven by competitive incentives at current capability levels (Kulveit et al., 2025). Preference-based alignment approaches (Ziegler et al., 2020; Ouyang et al., 2022; Bai et al., 2022a;b) have substantially improved model behavior. Evaluation and red-teaming work has exposed persistent failure modes including sycophancy, specification gaming, and reward-tampering risks (Perez et al., 2023; Denison et al., 2024; Sharma et al., 2024). Work on structural limits of alignment (Cao, 2025a) shows that bounded evaluator capacity constrains the alignment process, a structural constraint that also reduces surface discriminability between human and AI outputs. Reward-model overoptimization (Gao et al., 2023) demonstrates how improving proxy metrics can diverge from true objectives, a dynamic that parallels value collapse when better-aligned outputs intensify adverse selection by narrowing the observable gap between production types. Model collapse research (Shumailov et al., 2024; Alemohammad et al., 2024) and its definitional scope (Schaeffer et al., 2025) examines recursive degradation from training on model-generated data. Our value-collapse mechanism identifies an economic channel that increases exposure to this technical risk.

**Governance Responses and Provenance Technologies.** YouTube (YouTube, 2024), Meta (Meta, 2025), TikTok (TikTok, 2024), and Spotify (Spotify, 2025) have introduced AI content labeling requirements. EU regulatory frameworks for marking AI-generated content (European Commission, AI Office, 2025) represent emerging governance infrastructure. Major publishers have established AI disclosure policies (Gewaltig, 2025; Zhu, 2025), and courts have imposed standing orders and sanctions (United States District Court for the Southern District of New York, 2023; United States District Court for the Eastern District of Pennsylvania, 2023). Watermarking (Kirchenbauer et al., 2023; Zhao et al., 2024) and detection methods underpin many of these efforts, though adversarial dynamics pose persistent challenges (Sadasivan et al., 2025). In our framework, governance measures and provenance technologies aim to raise verification ability or reduce per-item inspection costs.

## 8. Call to Action

**Make provenance observable.** Provenance systems, authenticated workflows, disclosure backed by audit mechanisms, and watermarking where robust can all make the production process more transparent. Because adverse selection operates through hidden provenance, making production visible directly counteracts the mechanism that drives undifferentiated evaluation. Voluntary disclosure faces a standard incentive problem. Producers who benefit from undifferentiated evaluation have no reason to reveal low-HTL provenance. Disclosure becomes informative only when tied to audits, liability exposure, procurement rules,

or platform mandates. Institutional precedents are emerging in legal sanctions, publishing disclosure requirements, and platform labeling rules (United States District Court for the Southern District of New York, 2023; European Commission, AI Office, 2025; Gewaltig, 2025). In our framework, these measures raise verification ability $g$.

**Reduce verification costs.** When per-item verification is expensive relative to the quality stakes involved, even evaluators who recognize quality differences cannot justify inspection at production volume. Better verification tools, citation and artifact checks, reviewer support infrastructure, and audit sampling can reduce per-item costs. Transparency about synthetic content in training data would allow downstream evaluators to calibrate expectations, and reporting estimated fractions of AI-generated content with audit mechanisms would serve this function. In academic peer review, where reviewer labor is already strained at scale (Aczel et al., 2021), allocating explicit time for source verification addresses the capacity constraint that makes inspection uneconomical. Research on efficient cryptographic protocols and statistical signatures can enable more economical source authentication. In our framework, these measures lower per-item verification cost $c_v$.

**Reward HTL contributions.** Adverse selection penalizes high-cost production when rewards are blind to how outputs were produced. Counteracting this requires evaluation criteria that make HTL visible in reward allocation. Funding and evaluation that weight sustained research programs and cumulative contributions more heavily than raw publication counts would shift incentives toward extended engagement with problems. Institutions could incentivize the *learning process itself* by requiring hands-on experience that preserves situational judgment even as AI tools assist with routine tasks. Content platforms could invest in provenance systems that make human authorship signals visible in recommendation and monetization where economically feasible.

**Protect HTL pipelines.** Pipeline compression erodes the evaluator pool that sustains verification capacity. Protecting junior training pipelines, apprenticeship structures, and long-term research programs directly addresses this channel. Maintaining experiential requirements where junior researchers engage in extended problem-solving preserves the pathway through which senior judgment develops. Investigating training procedures that preserve distributional diversity under synthetic data mixing (Shumailov et al., 2024; Alemohammad et al., 2024) can limit recursive degradation. Monitoring HTL participation trends would provide early warning before erosion becomes difficult to reverse. Relevant indicators include enrollment in multi-year training programs, early-career employment in HTL-intensive

roles (Brynjolfsson et al., 2025a; SignalFire, 2025), and quality metrics across knowledge-production domains. Sustained participation in HTL-intensive production maintains the evaluator pool on which provenance systems, verification tools, and HTL-sensitive rewards all depend. Qualified evaluators emerge from extended engagement with the problems they are asked to judge.

## 9. Conclusion

Our framework shows that value collapse is already operating across knowledge and cultural production through ordinary market selection, without requiring AGI-level capabilities or alignment failures. Better alignment and greater capability intensify this pressure by narrowing the observable gap between human and AI outputs. As deep human work exits the pool, the evaluator base that sustains verification erodes, and the feedback signal on which alignment depends degrades. Preserving the human judgment that effective AI safety requires means rewarding the *learning process itself*.

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

# A. A Reduced-Form Costly-Inspection Model

This appendix provides a reduced-form costly-inspection formalization underlying the analysis in the main text. Our mechanism is closest to Akerlof's quality-uncertainty model (Akerlof, 1970). When source-sensitive inspection is not worthwhile, rewards pool across heterogeneous production types and high-cost producers face exit pressure. We use a simple costly-inspection formulation that supports stage comparison across domains. Related work on adverse selection (Rothschild & Stiglitz, 1976) and signaling (Spence, 1973) provides broader context.

### A.1. Players, Types, and Timing

**Players.** $P$ (producer) and $B$ (buyer/decision maker). In applied settings, $B$ may be a journal editor, hiring committee, content platform, funder, or any agent allocating rewards or access based on output quality.

**Types.** The producer's type $\theta \in \{H, L\}$ is private information.

- Type $H$: HTL-intensive production, involving extended temporal learning.

- Type $L$: AI-primary or low-HTL production, relying primarily on generative tools.

**Prior.** $\Pr(\theta = H) = \lambda$, $\Pr(\theta = L) = 1 - \lambda$, where $\lambda \in [0, 1]$ denotes the buyer's prior over submitted outputs in period $t$, shaped by prior exit decisions. Each active producer submits one output per period.

**Costs and Qualities.** Expected output values satisfy $q_H > q_L$, where $q_\theta$ denotes the buyer's monetary valuation of type-$\theta$ output under full information. The quality gap is $\Delta q = q_H - q_L > 0$. Production costs are denoted $c_\theta$ for type $\theta$, with $c_H \gg c_L$, reflecting the time-intensive nature of HTL. We assume $c_L \leq q_L < c_H < q_H$. The condition $q_H > c_H$ keeps HTL-intensive production viable under full-information rewards, and the condition $q_L \geq c_L$ keeps low-HTL production active under low-type valuation. Pooled rewards can fall below $c_H$ when $\lambda$ is small, because $q_L < c_H$ ensures that a pool dominated by low-HTL output cannot sustain HTL-intensive producers.

**Timing.**

1. Nature draws $\theta$.

2. Producer observes $\theta$ and chooses *enter* or *exit*. If exit, producer receives payoff 0.

3. If enter, the producer generates and submits output.

4. Buyer chooses *inspect* at cost $c_v > 0$ or *not inspect*.

5. If the buyer inspects, the inspection procedure yields a signal about $\theta$ with effective informativeness $g$. If the buyer does not inspect, no provenance information is obtained beyond the pool prior.

6. Buyer assigns reward or allocation based on available information.

7. Payoffs are realized.

### A.2. Inspection Technology

The parameter $g \in [0, 1]$ denotes the effective informativeness of the feasible inspection procedure. When the buyer inspects at cost $c_v$, the inspection yields decision-relevant information about $\theta$ with informativeness $g$. Higher $g$ means the inspection more reliably distinguishes type $H$ from type $L$. When $g = 0$, inspection yields no useful provenance information. When $g = 1$, inspection perfectly reveals $\theta$. We treat $g$ as a reduced-form index without specifying a particular signal structure such as binary revelation or a continuous noisy signal, since the main-text analysis requires only the monotonicity properties described below.

## A.3. Gross Benefit of Inspection and Verification Condition

Let $V(g, \lambda, \Delta q)$ denote the gross benefit of inspection, defined as the expected improvement in the buyer's allocation payoff from inspecting relative to deciding based solely on the pool prior $\lambda$. We assume:

- $V$ is increasing in $g$ (more informative inspection yields better allocation decisions);

- $V$ is increasing in $\Delta q$ (larger quality stakes make correct classification more valuable);

- $V$ depends on pool composition $\lambda$ (the value of resolving type uncertainty varies with the mix of producers).

**Proposition A.1** (Inspection Condition)**.** *The buyer inspects if and only if the gross benefit of inspection is at least the inspection cost,*

$$V(g, \lambda, \Delta q) \geq c_v. \tag{4}$$

$V$ denotes the evaluator's private benefit from inspection. Downstream externalities of HTL erosion, including degraded training data, a shrinking evaluator pool, and reduced capacity for novel problem-solving, may make the social value of inspection considerably larger than the private value. When this gap is wide, evaluators rationally forgo verification even when inspection would be socially worthwhile, providing the economic rationale for governance interventions aimed at raising verification ability or lowering inspection costs.

For the main-text cross-domain comparison, composition effects are absorbed into a normalized reduced-form, yielding the threshold $g^* \approx c_v/\Delta q$. A convenient separable specification is $V(g, \lambda, \Delta q) = g \cdot h(\lambda) \cdot \Delta q$, where $h(\lambda)$ captures how pool composition affects the value of resolving type uncertainty under the domain's allocation rule. Setting $h(\lambda) = 1$ is a simplification adopted for cross-domain comparison in the main text. It suppresses the effect of pool composition, isolating the roles of inspection informativeness and quality stakes. The full condition retains $\lambda$ through $h(\lambda)$, which matters when analyzing within-domain dynamics where pool composition shifts over time.

## A.4. Payoffs

**Buyer.** Inspection improves the probability of correct allocation. The buyer benefits from accepting HTL-intensive work that merits acceptance, discounting AI-primary work where quality falls short, and avoiding costly mistakes such as publishing fabricated content, deploying flawed code, or filing erroneous legal briefs. The buyer's payoff from better allocation is proportional to the quality gap $\Delta q$ modulated by inspection informativeness. Under non-inspection, the buyer evaluates based on pooled expected quality and bears the cost of misallocation.

**Producer.** If the producer exits, payoff is $0$. If the producer enters under pooled evaluation, payoff is $\bar{p}_t - c_\theta$, where $\bar{p}_t$ is the pooled reward at time $t$ and $c_\theta$ is the type-specific production cost defined above. If source-sensitive verification is feasible, type-dependent reward may reflect $q_\theta$.

## A.5. Non-Inspection and Pooled Evaluation

**Proposition A.2** (Non-Inspection and Participation)**.** *If $V(g, \lambda, \Delta q) < c_v$, the buyer does not inspect. Under non-inspection, rewards track pooled expected quality,*

$$\bar{p}_t = \lambda_t q_H + (1 - \lambda_t)q_L. \tag{5}$$

*Type H enters if and only if $\bar{p}_t \geq c_H$. Type L enters if and only if $\bar{p}_t \geq c_L$. If $\bar{p}_t < c_H$ and $\bar{p}_t \geq c_L$, type H exits while type L remains.*

## A.6. Exit Dynamics Under Pooled Evaluation

**Proposition A.3** (Decline in $\lambda$)**.** *Assume that type-H producers exit whenever $\bar{p}_t < c_H$, while type-L producers remain active whenever $\bar{p}_t \geq c_L$. With a single representative cost $c_H$, exit is simultaneous. With heterogeneous costs within type H, exit is progressive and the unraveling dynamic strengthens. When costs are heterogeneous, $c_H$ represents the marginal exit threshold rather than a single cost level. Because the assumption $q_H > c_H$ keeps HTL-intensive production viable under full-information pricing, we have $\bar{p}_t < c_H \leq q_H$, confirming that pooled pricing falls short of what type-H output is worth. Combined with $\bar{p}_t \geq c_L$, this implies $\lambda_{t+1} < \lambda_t$. As $\lambda_t$ falls, $\bar{p}_t = \lambda_t q_H + (1 - \lambda_t)q_L$ falls, and each further exit lowers pooled reward, triggering additional exit. Under continued non-inspection, $\lambda_t$ can converge to $0$ and $\bar{p}_t$ to $q_L$.*

The quality of the pool degrades as HTL-intensive producers exit, which further lowers the pooled reward and can trigger additional exit when costs are heterogeneous. The market continues to function, with outputs still produced and evaluated, but the composition shifts systematically away from HTL-intensive work. Under the stylized assumptions, once $\lambda$ approaches zero, the pool remains in the low-HTL state and does not recover without external intervention. Institutional subsidies, reputation premia, licensing requirements, or intrinsic motivation may sustain positive HTL participation in practice.

The model identifies the conditions under which adverse-selection pressure operates. The four-stage framework in Section 4 maps domains according to how strongly these conditions hold.

### A.7. Relation to the Main Text

The main text uses the simplified threshold $g^* \approx c_v/\Delta q$ for accessibility and cross-domain comparison. This appendix provides the full condition $V(g, \lambda, \Delta q) \geq c_v$, retaining the dependence on pool composition. The simplified threshold highlights the role of inspection informativeness and quality stakes. The full condition additionally accounts for the current HTL share $\lambda$. Both formulations yield the same comparative statics with respect to $g$, $c_v$, and $\Delta q$ when composition effects are held fixed. Holding $g$ and composition fixed, domains with high $c_v$ relative to $\Delta q$ are most vulnerable to non-inspection and adverse selection. Under the full condition, vulnerability depends on $c_v$ relative to $g\,h(\lambda)\,\Delta q$.

