# OpenReview forum: "Position: Generative Models Erode Human Temporal Learning Through Market Selection"
_ICML.cc/2026/Position_Paper_Track — ICML 2026 Position Paper Track regular_

### Official Review · Reviewer_uNia · 2026-02-14

**Significance:** 3
**Argument Clarity:** 2
**Rating:** 3
**Confidence:** 3

**Questions:**

1. Can you use an explicit model, with some formulas, to describe the competition between temporal learning and using AI? You may use Nash Equilibrium to describe market behavior and show why temporal producers exit.

2. For "Major model providers should publicly report estimated fractions of AI-generated
content in training data, establishing transparency
around data provenance", we don't have a reasonable mechanism to motivate humans to report. Can you add more discussion on how to incentivize people to do that?

**Alternative Views Section:**

Yes

**Compliance With Llm Reviewing Policy A Conservative:**

Affirmed.

**Discussion Potential:**

2

**Final Justification:**

In my view, the authors fail to connect real-world data to justify why the assumptions in the paper are reasonable. At the same time, some claims, such as “In NHANES, expert audit costs ~1,000+ USD per paper while the marginal scientific value of redundant single-factor studies approaches zero”, are questionable. Therefore, I am inclined to maintain my score after reading the judgments of the other reviewers.

**Paper Summary:**

The paper discusses an important question: whether generative models, before reaching the level of AGI, may affect human temporal learning. By analyzing market selection mechanisms, the authors show that the market may automatically crowd out temporal learning, ultimately leading to a value collapse.

**Position:**

Yes

**Position In Title:**

Yes

**Related Work:**

3

**Strengths And Weaknesses:**

The topic discussed in this paper is highly important and has recently received widespread attention from the community.

The paper’s arguments lack sufficient empirical support. For example, the claim that “the critical threshold occurs when verification costs exceed expected benefits” would be much more convincing if the authors could provide estimates of these costs and thresholds. Similarly, statements, such as “Analysis of millions of abstracts shows researchers using LLMs substantially increased publication output following ChatGPT’s release. By recent estimates, significant fractions of computer science papers and peer reviews show evidence of LLM usage (Liang et al., 2025)", require more concrete data, such as quantitative proportions or specific estimates, to substantiate these claims.

**Support:**

2

---

> ### Author Rebuttal · Authors · 2026-03-31
>
> Thank you for your review! We address your concerns in three ways: concrete quantitative anchors replacing vague claims, an explicit Bayesian screening game, and institutional disclosure incentives.
>
> >**W1: The paper's arguments lack sufficient empirical support.**
>
> We agree. Vague formulations are now replaced with specific anchors: LLM modification in CS abstracts reaches 22.5%, up from 2.4% pre-ChatGPT [1]; NHANES formulaic publications surged ~47-fold [2]; early-career workers (ages 22–25) in AI-exposed occupations experienced 16% relative employment decline while ages 35–49 saw 6–9% growth [3]; U.S. law graduates maintained 93.4% employment [4].
>
> The parameter $g$ cannot be directly measured, but its decline has three independent lines of support:
>
> 1. AI training minimizes divergence between model and human output distributions, so AI training puts downward pressure on $g$;
> 2. Liang et al. report that "reliably identifying LLM-generated or LLM-modified content at the individual level remains a complex and unresolved task" [1];
> 3. screening outcomes are consistent: fabricated citations pass expert review [5], formulaic papers enter literature at unprecedented volume [2].
>
> The threshold $g^\ast$ can be bounded without precise parameter values. For NHANES, the marginal scientific value of redundant single-factor papers is often very low while per-paper verification can easily reach the order of USD 1,000 per paper, placing $g^\ast$ well above 1, predicting unconstrained entry, matching the observed surge. For legal services, error costs in representative cases exceed verification costs by orders of magnitude, so $g^\ast$ is well below 1, predicting continued screening, matching employment stability. The core claim is therefore falsifiable: domains with lower screening payoff relative to verification cost should enter pooled evaluation earlier, and the current cross-domain ordering is consistent with that prediction. NHANES offers a suggestive contrast: dataset and methods held relatively constant while production technology changed.
>
> >**Q1: Can you use an explicit model, with some formulas, to describe the competition between temporal learning and using AI?**
>
> We state the screening game explicitly. Producers are of type $\theta \in {H, L}$, incurring costs $c_H \gg c_L$ and producing expected qualities $q_H > q_L$. A buyer observes surface features with discriminative signal $g$ but not type. Screening at cost $c_v$ yields expected benefit $g \cdot \Delta q$ ($\Delta q = q_H - q_L$). In equilibrium, if $g \cdot \Delta q < c_v$ and pooled price $\bar{p} < c_H$, buyers do not screen, type-$H$ exits, type-$L$ enters. No player benefits from unilateral deviation. As the high-HTL share $\lambda$ falls, $\bar{p}$ declines, following Akerlof unraveling, absorbing at $\lambda=0$. This is a Bayesian Nash Equilibrium since buyers cannot observe producer type; the revised manuscript gives the full specification. Temporal producers exit because pooled prices cannot cover HTL production costs.
>
> >**Q2: For 'Major model providers should publicly report estimated fractions of AI-generated content in training data', we don't have a reasonable mechanism to motivate humans to report.**
>
> We do not rely on voluntary honesty; disclosure becomes incentive-compatible only when tied to audits, liability, procurement rules, safe harbors, or platform/publisher mandates. Those who most need to report face the strongest counter-incentive, since disclosure would reduce competitive advantage. Regulatory enforcement is already underway: courts have already sanctioned AI-fabricated filings [6]; major publishers require AI disclosure [7]; the EU AI Act mandates foundation model transparency; YouTube and TikTok require AI content labeling. Enforcement varies with market type: small expert communities can self-govern through norms; large anonymous markets require regulatory frameworks.
>
> **References**
>
> [1] Quantifying Large Language Model Usage in Scientific Papers. Nature Human Behaviour, 2025
>
> [2] Explosion of Formulaic Research Articles, Including Inappropriate Study Designs and False Discoveries, Based on the NHANES US National Health Database. PLoS Biology, 2025
>
> [3] Canaries in the Coal Mine? Six Facts about the Recent Employment Effects of Artificial Intelligence. Stanford Digital Economy Lab Working Paper, 2025
>
> [4] Employment Summary for the Class of 2024. NALP Annual Report, 2025
>
> [5] GPTZero Finds Over 50 New Hallucinations in ICLR 2026 Submissions. GPTZero Investigations, 2025
>
> [6] Mata v. Avianca, Inc.: Opinion and Order on Sanctions. Southern District of New York, 2023
>
> [7] Generative AI for Content Discovery in Academic Publishing: A Content Provider’s Perspective. Information Services and Use, 2025

---

> > ### Author Rebuttal · Reviewer_uNia · 2026-04-03
> >
> > Can the authors discuss the direct connection between the newly added data and their claims? For example, how does the magnitude of $g^\ast$ vary across different industries?

---

### Official Review · Reviewer_eNf8 · 2026-03-14

**Significance:** 4
**Argument Clarity:** 3
**Rating:** 5
**Confidence:** 4

**Questions:**

- One response to the paper, or corollary of the paper may be we should directly incentivize temporal learning not for the output product (since they will be mimiced by the machines in the future) but for the learning and training process itself — since training the human beings still have immense value for us. We should write papers to facilitate learning in ourselves instead of just knowledge creation.
- Another response to the paper is like, is this logic super different from some other domains e.g. textile industry, where it was human doing it for a while, before machine produced clothes taking over using its much cheaper price and much larger production rate. It has happened 100 years ago, where manual textile worker (e.g. in China ~ 1900) got squeezed severely due to the import of machine produced clothes. But still 100 years later, there are much fewer manual textile worker, but they will produce high value handwoven clothes themselves.

**Alternative Views Section:**

Yes

**Compliance With Llm Reviewing Policy A Conservative:**

Affirmed.

**Discussion Potential:**

4

**Final Justification:**

I'd keep the accept score as is. The response of the authors has satisfactorily addressed all my concerns about writing clarity and the conceptual points.
We applaude the authors for a relevant and elegant position paper.

**Paper Summary:**

This essay provides a very engaging and eloquent argument towards the undesirable economic effects of AI generated content. i.e. the cheap production of near identical product from machines will take a much larger market share, squeezing away the space of human temporal learning (years of experience and training). The value of product will decrease due to machine production, leading to positive feedback loop. This mechanism is neglected by the traditional AI superpower taking over narrative and the AI alignment research. This risk is on the market level, not on the individual level. One key statement I feel insightful is “*The mechanism activates when generators approximate output distributions closely enough that expected economic gain from screening becomes small relative to verification cost*”, which is quite relevant and timely.

**Position:**

Yes

**Position In Title:**

Yes

**Related Work:**

3

**Strengths And Weaknesses:**

**Strength**

- The usage of term temporality and comment on temporal learning in this context is quite novel and illuminative. Indeed a big difference between human expert and generative model is that human requires our own training and learning process.
- The paper has a moderate view of the issue, not exaggerating it to be eye catching, but keeping a reasonable voice.
- The argument is concise but convincing, many of us can feel the relevance and truth in it. It’s also novel and indeed orthogonal for some other account of AI risk.

**Weakness**

- In some part the language is a bit philosophical and poetic, but it’s a style preference, not a critical weakness.
    - For examples, I’m not sure the usage of the phrase "temporal work", "temporal learning", adds much in the article, seems many of them can be substituted by "human work", "human learning"? — which may make the article less mysterious. The term "temporal learning" is somewhat ambiguous in the title and common ML reader may think it means something else (e.g. learning temporal sequence), but "human learning" or "human temporal learning" can be more unambiguous.
- The use of “*Representation learning and autoregressive generation*” in several part of the article is quite strange, not sure why and how these two terms can be put together as parallel… May be author wants to say generative modeling in general, since autoregressive generation is not the only way for generation (diffusion and flow as the other one). Or maybe the author mean supervised learning instead of representation learning.

    > e.g. *“ Representation learning and autoregressive generation (Bengio*
    *et al., 2014; Vaswani et al., 2017) treat tasks as fitting*
    *generators over recorded outputs, omitting slow human*
    *learning that produced them.”*
    >

**Support:**

3

---

> ### Author Rebuttal · Authors · 2026-03-31
>
> Thank you for your generous assessment and the insights that strengthened the paper!
>
> >**W1(a): In some part the language is a bit philosophical and poetic, but it's a style preference, not a critical weakness.**
>
> We appreciate the understanding. The revision makes the screening mechanism more explicit and adds a cross-domain calibration.
>
> >**W1(b): The term "temporal learning" is somewhat ambiguous in the title and common ML reader may think it means something else (e.g. learning temporal sequence), but "human learning" or "human temporal learning" can be more unambiguous.**
>
> We agree. The revision uses **Human Temporal Learning (HTL)** throughout, with explicit clarification: "not to be confused with learning temporal sequences in ML."
>
> >**W2: The use of “Representation learning and autoregressive generation” in several part of the article is quite strange, not sure why and how these two terms can be put together as parallel.**
>
> We agree this pairing was imprecise. The revision uses **generative modeling broadly construed** as the umbrella term, with autoregressive, diffusion, and flow-based approaches discussed where technically relevant.
>
> >**Q1: One response to the paper, or corollary of the paper may be we should directly incentivize temporal learning not for the output product (since they will be mimiced by the machines in the future) but for the learning and training process itself — since training the human beings still have immense value for us. We should write papers to facilitate learning in ourselves instead of just knowledge creation.**
>
> This suggestion aligns closely with our analysis. A central implication of the framework is that output-based reward mechanisms lose discriminative power as AI replicates surface features of HTL-intensive outputs. The revised call to action incorporates this direction: weighting cumulative research programs over publication counts, and designing experiential requirements that preserve situational learning under AI-assisted workflows.
>
> >**Q2: Another response to the paper is like, is this logic super different from some other domains e.g. textile industry, where it was human doing it for a while, before machine produced clothes taking over using its much cheaper price and much larger production rate. It has happened 100 years ago, where manual textile worker (e.g. in China ~ 1900) got squeezed severely due to the import of machine produced clothes. But still 100 years later, there are much fewer manual textile worker, but they will produce high value handwoven clothes themselves.**
>
> We thank the reviewer for this helpful analogy. The prediction may be partially correct: HTL-intensive work could retreat to a high-end niche, much as handwoven textiles survived industrialization.
>
> The societal consequences, however, differ in a structural way. Textiles are terminal consumption goods: machine-produced cloth does not re-enter the loom's design process. When handweaving retreated, machine textiles served consumers without any input from artisans, and externalities were minimal.
>
> Knowledge is a production factor. Today's research is the basis for tomorrow's research and the training data for next-generation models. If HTL retreats to a niche, training data diversity declines, the evaluator pool contracts, and society's capacity to respond to novel problems degrades. These are diffuse externalities borne by the knowledge commons. Additionally, AI-generated content re-enters training corpora, creating a recursive feedback loop with no analogue in physical manufacturing. The revision discusses this pathway in Section 5.3.

---

> > ### Author Rebuttal · Reviewer_eNf8 · 2026-04-01
> >
> > The authors answers all our concern about the writing style and content in a satisfactory way!
> > We believe the final version will be more elegant and impactful to the audience.

---

### Official Review · Reviewer_8A9z · 2026-03-16

**Significance:** 4
**Argument Clarity:** 3
**Rating:** 5
**Confidence:** 4

**Questions:**

Q1. Can the authors address weakness W2? Are there examples of value collapse or the propagation of LLM content from settings where there is a considerable cost to the user of unverified material? If there are not clear-cut examples, what are the main components of the argument that the value collapse will extend to these domains? (while I might use an LLM for accounting guidance to help make some decisions, I will pay an accountancy firm, with verified temporal learning evidence through degrees and prior work, to prepare company tax filings, because there is a very high cost associated with false or erroneous filings).

**Alternative Views Section:**

Yes

**Compliance With Llm Reviewing Policy A Conservative:**

Affirmed.

**Discussion Potential:**

4

**Final Justification:**

The authors provided a thorough rebuttal to the main criticisms in my review and as a result I have increased the recommendation to "accept". The rebuttal does promise multiple revisions (both in response to the issues I raised and those raised by other reviewers) and a "strong accept" recommendation would require a review of the revised paper.

**Paper Summary:**

The paper takes the position that machine learning (generative AI in particular) poses major risks for knowledge and cultural production. The claim is that the value of temporality (the time invested in a human learning or acquiring expertise) is likely to diminish or even disappear as machines become increasingly capable of generating content that cannot be meaningfully distinguished from human-generated content without a costly audit process. The paper refers to this as “value collapse” and motivates the position using examples from academic publishing.

**Position:**

Yes

**Position In Title:**

Yes

**Related Work:**

3

**Strengths And Weaknesses:**

S1. The paper makes a clear case that the position is timely and important. The paper draws attention to societal risks associated with the position. There is a high probability that the position will garner interest and lead to lively, positive discussion.

S2. The position is clearly and consistently articulated and the paper conforms to the requirements of a position paper, including a thorough discussion of potential alternative viewpoints.

S3. The paper provides a detailed explanation of the position and associated argumentative support, explaining why temporality has (or leads to) economic value (Section 2) and how generative modeling can act as a mechanism that erodes that value (Sections 3 and 4). Section 5 brings together the arguments and provides examples of setting where there is preliminary evidence that can be interpreted as supporting the position.

W1. The paper does not sufficiently outline some key assumptions that underpin the presented position. First, the presented analysis assumes that there are only two outputs to choose between, i.e., AI-generated or human-expert (reflecting temporal learning). In practice, many experts are already making use of AI, leading to products that are a hybrid of the two. Although the paper acknowledges this to a limited extent in the Alternative Views section, it is more important that it is stated up-front as a core assumption. The presented economic analysis does not apply without it.  Second, the analysis assumes that the only option for a “buyer” becomes verification via screening. It is not clearly established why “doctoral training, publication records, and demonstrated mastery” can be used as credible indicators of expertise now but cannot be used in the future.

W2. The paper’s examples (academic publishing, digital platforms) do not clearly extend to the many settings where there are considerable costs associated with propagating, producing or using false content. These selected example settings were subject to fraudulent or plagiarized contributions before the advent of LLMs (much of the digital content on many platforms in 2015 was unapproved copies and edits of copyrighted material). Monitored platforms like Netflix do not contain any more generated LLM content today than they did 5 years ago. While LLMs may have (greatly) exacerbated these issues in these specific domains, the paper does not make a compelling case that the identified problems (the value collapse) will extend to scenarios where it is important or even vital to have validated work from an authorized source.

**Support:**

3

---

> ### Author Rebuttal · Authors · 2026-03-31
>
> Thank you for your careful review! We clarify two points: the paper assumes hybrid production, and value collapse is domain-heterogeneous rather than universal.
>
> >**W1(a): The paper does not sufficiently outline some key assumptions that underpin the presented position. First, the presented analysis assumes that there are only two outputs to choose between.**
>
> We agree this needed to be stated up front. We now make this assumption explicit: both producer types use AI tools, differing in irreducible human temporal investment. In reality this is a continuum; the formal model compresses it to two types for tractability.
>
> >**W1(b): Second, the analysis assumes that the only option for a 'buyer' becomes verification via screening.**
>
> We do not claim they stop mattering. They certify capacity for deep work, and remain useful producer-level priors. What they cannot do is certify whether that capacity was exercised in a specific output. A credentialed researcher may invest substantial HTL in one paper and rely heavily on AI in the next.
>
> Our deeper concern is that AI inverts the default mode of production. Previously, careful human work was the default; cutting corners required deliberate effort. Now, generating output with minimal temporal investment is the path of least resistance. The credential persists, but the probability it guarantees—that a given output reflects deep engagement—may be quietly declining. Because adverse outcomes remain rare in high-penalty domains, this erosion can proceed undetected. We do not claim all trust mechanisms collapse into per-item screening. In authorized-source or centrally controlled settings, producer-level priors and provenance systems may remain effective for much longer; the concern is narrower: it arises when these mechanisms cannot certify whether a marginal output reflects substantial Human Temporal Learning, and per-item verification remains too costly relative to the gain from distinguishing sources.
>
> >**W2: The paper’s examples (academic publishing, digital platforms) do not clearly extend to the many settings where there are considerable costs associated with propagating, producing or using false content.**
> >
> >**Q1: Can the authors address weakness W2? Are there examples of value collapse or the propagation of LLM content from settings where there is a considerable cost to the user of unverified material? If there are not clear-cut examples, what are the main components of the argument that the value collapse will extend to these domains?**
>
> Our model predicts the pattern the reviewer describes. The revision introduces a screening threshold $g^\ast = c_v / \Delta q$, the ratio of verification cost to quality stakes: where quality stakes are large, $g^\ast$ is low and screening persists longer. Value collapse is reframed as a process with observable stages:
>
> **Stage 1** (AI enters workflow, screening intact): AI-assisted clinical documentation and summarization are useful in practice, but physician review remains necessary [1].
>
> **Stage 2** (Verification burden rises, institutional overlays appear): Courts sanction attorneys for AI-fabricated citations [2,3] and add certification requirements [4]—screening succeeding at increasing cost.
>
> **Stage 3** (Volume overwhelms verification): NHANES publications surge ~47-fold [5]; fabricated citations pass 3–5 ICLR reviewers [6].
>
> **Stage 4** (Pooled evaluation): Content platforms.
>
> High-penalty domains occupy Stages 1–2, as predicted. We also note evidence from software security: the cURL maintainer reports ~20% of 2025 security submissions were AI slop, each requiring 3–4 experts for up to 3 hours, with declining valid-report rates, leading to consideration of ending the bug bounty [7]. This illustrates the mechanism in a domain where $\Delta q$ is unambiguously large.
>
> On the accountant example: accounting has very high $\Delta q$, yielding low $g^*$. The framework predicts current stability. The longer-term concern is junior pipeline compression: if AI automates entry-level tasks, the experiential path building senior judgment may narrow while final outputs remain screened.
>
> We appreciate these points; they materially improved the paper.
>
> **References**
>
> [1] Generative Artificial Intelligence to Transform Inpatient Discharge Summaries to Patient-Friendly Language and Format. JAMA Network Open, 2024
>
> [2] Mata v. Avianca, Inc.: Opinion and Order on Sanctions. Southern District of New York, 2023
>
> [3] In re: Eric Chibueze Nwaubani, No. 25-9517. Fourth Circuit, 2026
>
> [4] Standing Order Re: Artificial Intelligence (“AI”) in Cases Assigned to Judge Baylson. Eastern District of Pennsylvania, 2023
>
> [5] Explosion of Formulaic Research Articles, Including Inappropriate Study Designs and False Discoveries, Based on the NHANES US National Health Database. PLoS Biology, 2025
>
> [6] GPTZero Finds Over 50 New Hallucinations in ICLR 2026 Submissions. GPTZero Investigations, 2025
>
> [7] Death by a Thousand Slops. Daniel Stenberg Blog, 2025

---

> > ### Author Rebuttal · Reviewer_8A9z · 2026-04-03
> >
> > The rebuttal resolves most of my concerns, and accordingly, I have raised my recommendation to "accept". To go beyond that, I would need to review the revised paper that implements the promised changes in response to my criticisms and those of the other reviewers.

---

### Decision · Program_Chairs · 2026-04-30

**Decision:**

Accept (regular)

**Comment:**

Reviewers found the paper timely, clear, and conceptually novel, particularly in connecting machine learning systems with economic mechanisms. While one reviewer raised concerns about empirical grounding and provided a lower score, the authors’ rebuttal addressed these issues by adding quantitative anchors and clarifying assumptions. Overall, this is a well-written, well articulated position paper.